# An observational analysis of the trope "A p-value of < 0.05 was considered statistically significant" and other cut-and-paste statistical methods

**Nicole M. White**[1]*, **Thirunavukarasu Balasubramaniam**[2], **Richi Nayak**[2], **Adrian G. Barnett**[1]

**1** Australian Centre for Health Services Innovation and Centre for Healthcare Transformation, School of Public Health and Social Work, Queensland University of Technology, Brisbane, Australia, **2** Centre for Data Science, School of Computer Science, Queensland University of Technology, Brisbane, Australia

* nm.white@qut.edu.au

## Abstract

Appropriate descriptions of statistical methods are essential for evaluating research quality and reproducibility. Despite continued efforts to improve reporting in publications, inadequate descriptions of statistical methods persist. At times, reading statistical methods sections can conjure feelings of *dèjá vu*, with content resembling cut-and-pasted or "boilerplate text" from already published work. Instances of boilerplate text suggest a mechanistic approach to statistical analysis, where the same default methods are being used and described using standardized text. To investigate the extent of this practice, we analyzed text extracted from published statistical methods sections from PLOS ONE and the Australian and New Zealand Clinical Trials Registry (ANZCTR). Topic modeling was applied to analyze data from 111,731 papers published in PLOS ONE and 9,523 studies registered with the ANZCTR. PLOS ONE topics emphasized definitions of statistical significance, software and descriptive statistics. One in three PLOS ONE papers contained at least 1 sentence that was a direct copy from another paper. 12,675 papers (11%) closely matched to the sentence "a p-value < 0.05 was considered statistically significant". Common topics across ANZCTR studies differentiated between study designs and analysis methods, with matching text found in approximately 3% of sections. Our findings quantify a serious problem affecting the reporting of statistical methods and shed light on perceptions about the communication of statistics as part of the scientific process. Results further emphasize the importance of rigorous statistical review to ensure that adequate descriptions of methods are prioritized over relatively minor details such as p-values and software when reporting research outcomes.

## Introduction

An ideal statistical analysis uses appropriate methods to draw insights from data and inform the research questions. Unfortunately many current statistical analyses are far from ideal, with

**Data Availability Statement:** All data and code used for analysis are available on GitHub at https://github.com/agbarnett/stats_section.

**Funding:** AGB was funded by a National Health and Medical Research Council Fellowship (APP1117784); https://www.nhmrc.gov.au. The funders had no role in study design, data collection and analysis, decision to publish, or preparation of the manuscript.

**Competing interests:** The authors have declared that no competing interests exist.

researchers often using the wrong methods, misinterpreting the results, or failing to adequately check their assumptions [1]. Some researchers take a "mechanistic" approach to statistics, copying the few methods they know regardless of their appropriateness, and then going through the motions of the analysis [2]. Applying this form of methodological illiteracy is at odds with the principles of scientific inquiry, yet continues to pervade published scientific research [3]. This paradox has been exemplified during the COVID-19 pandemic, which has led to unprecedented levels of published research of largely poor quality [4, 5].

Many researchers lack adequate training in research methods, and statistics is something they do with trepidation and even ignorance [6, 7]. However, using the wrong statistical methods can cause real harm [6, 8] and bad statistical practices are being to used abet weak science [2]. Statistical mistakes are a key source of research waste and are contributing to the current reproducibility crisis in science [9]. Even when the correct methods are used, many researchers fail to describe them adequately, making it difficult to reproduce the results [10, 11]. Poor statistical methods might not be caught by reviewers, as they may not be qualified to judge the statistics. A recent survey of editors found that only 23% of health and medical journals used expert statistical review for all articles [12], which was little different from a survey from 22 years ago [13].

There is guidance for researchers on how to write up their statistical methods and results. The International Committee of Medical Journal Editors recommend that researchers should: "Describe statistical methods with enough detail to enable a knowledgeable reader with access to the original data to judge its appropriateness for the study and to verify the reported results" [14]. More detailed guidance is given by the SAMPL and EQUATOR guidelines [15, 16] covering all aspects of research reporting tailored to different study designs. Both of these guidelines were led by Doug Altman, who spoke for many years about the need for better statistical reporting. The awareness and use of these guidelines could be improved. There were 303 Google Scholar citations to the SAMPL paper (as at 8 October 2021) which is a good citation number for most papers, but is low considering the millions of papers that use statistical analysis.

A potential contributor to poor reporting is the temptation for researchers to re-use descriptions of the same default statistical methods, to make their papers resemble those of their peers and increase perceived chances of publication [17]. As these default choices become more common, valid criticism by reviewers and journal editors becomes increasingly difficult, as past use may be argued by researchers as offering precedent for the conduct of analysis within their discipline [18]. Two statisticians on this paper (AB and NW) have heard researchers admit that they have copied-and-pasted their statistical methods sections from other papers. To investigate the extent of this practice, we applied topic modelling to analyze text within statistical methods sections, as part of published journal articles and clinical trials protocols. Modelling results were used to estimate the extent that researchers are using cut-and-paste or 'boilerplate' statistical methods sections. Boilerplate text is that "which can be reused in new contexts or applications without significant changes to the original" [19]. The use of boilerplate text indicates that researchers are emphasizing the same details about chosen statistical analyses, and potentially giving little thought into the conduct and transparent reporting of statistical methods used.

## Materials and methods

### Data sources

We used two openly available data sources to find statistical methods sections: research articles published in *PLOS ONE* and study protocols registered on the Australian and New Zealand Clinical Trials Registry (ANZCTR). Data sources were chosen as examples of common

research outputs that include descriptions of statistical methods that are either planned or were used for analyzing studies.

**Public Library of Science (PLOS ONE).**   *PLOS ONE* is a open access mega-journal that publishes original research across a wide range of scientific fields. Article submissions are handled by an academic editor who selects peer reviewers based on their self-nominated area(s) of expertise. Currently there are 324 academic editors out of 9,648 (3%) with the keywords of "statistics (mathematics)" or "statistical methods" in their expertise list (web search on 25-May-2021, https://journals.plos.org/plosone/static/editorial-board). Submissions do not undergo formal statistical review. Instead, reviewers are required to assess submissions against several publication criteria, including whether: "Experiments, statistics, and other analyses are performed to a high technical standard and are described in sufficient detail" [20]. All reviewers are asked the question: "Has the statistical analysis been performed appropriately and rigorously?", with the possible responses of "Yes", "No" and "I don't know". In September 2019, author instructions were updated to allow citations of established materials, methods and protocols, provided sufficient details are given for approaches to be understood independently of chosen references [21]. Authors are encouraged to follow published reporting guidelines such as EQUATOR, to ensure that chosen statistical methods are appropriate for the study design, and adequate details are provided to enable independent replication of results.

Data on all *PLOS ONE* articles can be accessed via the PLOS Application Programming Interface (API). This enabled us to conduct searches of full-text articles and analyze data on articles' text content and general attributes such as publication date and field(s) of research. All available papers regardless of publication date were considered. We applied a two-step approach to identify statistical methods sections:

*Step 1*: Targeted API searches were completed using the R package 'rplos' [22]. Search queries targeted analysis-related terms, combining the words "data" or "statistical" with one of: "analysis", "analyses", "method", "methodology" or "model(l)ing". Terms could appear anywhere within the main body of the article, to account for the placement of relevant text in different sections, for example, in the *Material and Methods* section versus *Results*. Search results were indexed by a unique Digital Object Identifier (DOI). Attribute data collected per DOI included journal volume and subject classification(s).

*Step 2*: *PLOS ONE* does not use standardized headings to preface statistical methods sections. To address this, we performed partial matching on available headings against frequently used terms in initial search results: 'Statistical analysis', 'Statistical analyses', 'Statistical method', 'Statistical methods', 'Statistics', 'Data analysis' and 'Data analyses'. All available data were downloaded on 3 July 2020.

Code to complete steps 1 and 2 is available at https://github.com/agbarnett/stats_section/code/plosone.

**Australia and New Zealand Clinical Trials Registry (ANZCTR).**   The ANZCTR was established in 2005 as part of a coordinated global effort to improve research quality and transparency in clinical trials reporting; observational studies can also be registered. All studies registered on ANZCTR are publicly available and can be searched via an online portal (https://www.anzctr.org.au).

Details required for registration follow a standardized template [23], which covers participant eligibility, the intervention(s) being evaluated, study design and outcomes. The information provided must be in English. Studies are not peer reviewed.

For the statistical methods section, researchers are asked to provide a brief description of sample size calculations, statistical methods and planned analyses, although this section is not compulsory [23]. Studies are reviewed by ANZCTR staff for completeness of key information, which does not include the completeness of the statistical methods sections.

All studies available on ANZCTR were downloaded on 1 February 2020 in XML format. For our analysis, we used all text available in the "Statistical methods" section. We also collated basic information about the study including the study type (interventional or observational), submission date, number of funders and target sample size. These variables were chosen as we believed they might influence the completeness of the statistical methods section. For example, we hypothesized that larger studies and those with funding to be more complete. We were also interested in changes over time.

Studies prior to 2013 were excluded as the statistical methods section appeared to be introduced in 2013. Some studies were first registered on the alternative trial database *clinicaltrials.gov* and then also posted to ANZCTR. We excluded these studies because they almost all had no completed statistical methods section as this section is not included in *clinicaltrials.gov*.

## Statistical methods

**Full-text processing.** Text cleaning aimed to standardize notation and statistical terminology, whilst minimizing changes to article style and formatting. *R* code used for data extraction and cleaning is available from https://github.com/agbarnett/stats_section.

Mathematical notation was converted from Unicode characters to plain text. Symbols outside of Unicode blocks including '%' (percent) and '<' ('less-than') were converted into plain text. General formatting was removed, including carriage returns, punctuation marks, in-text references (e.g. "[42]") centered equations, and other non-ASCII characters. Bracketed text was retained with brackets removed to maximize content for analysis. Stop words including pronouns, contractions and selected prepositions were removed. We retained selected stop words that, if excluded, may have changed the context of statistical methods being described, for example 'between' and 'against'.

We compiled an extensive list of statistical terms to standardize reported descriptions of statistical methods. An initial list was compiled by calculating individual word frequencies and identifying relevant terms. Extra terms were sourced from index searches of three statistics textbooks [24–26]. Plurals (e.g., 'chi-squares') unhyphenated terms (e.g., 'chi square') and combined terms (e.g. 'chisquare') were transformed to singular, hyphenated form (e.g., 'chi-square'). Common statistical tests were also hyphenated (e.g., 'hosmer lemeshow' to 'hosmer-lemeshow').

**Analysis of missing statistical methods sections.** Statistical methods sections were missing for some studies downloaded from ANZCTR, including sections labelled as "Not applicable", "Nil" or "None". Since these studies would be excluded from topic modeling, we examined if there were particular studies where the statistical methods section was more likely to be missing. Analysis considered a logistic regression model estimated in the Bayesian framework ([27]; www.r-inla.org), with missing statistical methods section (yes/no) as the dependent variable. The independent variables were date, study type, number of funders and target sample size which was $\log_2$ transformed because of a large positive skew. Results were reported as odds ratios with 95% credible intervals.

**Topic modelling.** Text from statistical methods sections was analyzed using Non-Negative Matrix Factorization (NMF). NMF is an established approach for topic modelling, and provides an effective solution for text-based clustering when dealing with high-dimensional data [28, 29].

For $N$ studies, let $P \in R^{M \times N}$ denote a content matrix of text from statistical methods sections, comprising of $M$ unique terms. Text clustering algorithms for identifying common topics across studies requires $P$ to be represented with a vector space model. In our case, unique

terms in *P* are modelled using the tf-idf (term frequency × inverse document frequency) weighting schema, to account for the relative importance of common and rare terms.

A common problem facing text clustering algorithms is the curse of dimensionality due to the high number of terms in the doc × term matrix representation [30, 31]. Applying text-based methods based on distance, density or probability therefore face difficulties in high-dimensional settings [32–34]. Specifically, distances between near and far points becomes negligible [31]. This behavior directly affects the performance of distance-based clustering methods such as *k*-means [35] in accurately identifying subgroups (topics) present in the data. Furthermore, sparseness associated with high-dimensional matrix representations does not allow for differentiation between topics based on density differences [32, 36].

To address these limitations, NMF deals with high-dimensional data by mapping it to a lower-dimensional space. This mapping is achieved by approximating *P* with two factor matrices: $W \in R^{M \times g}$ and $H \in R^{N \times g}$ [31], such that $P \approx WH^T$. The number of subgroups of common topics inferred from the data is given by *g*.

The matrix factorization process approximates the lower dimensional non-negative factor matrices *W* and *H* such that they can represent high dimensional *P* with the least error. Estimation of *W* and *H* is achieved by optimizing an objective function; for NMF, the Fronbeius norm is used, equivalent to minimizing the sum of squares for all elements of *P*:

$$\min \frac{1}{2} \parallel P - WH \parallel = \sum_{i=1}^{M} \sum_{j=1}^{N} (P_{i,j} - (WH)_{i,j})^2$$

Following estimation, *H* contains the information regarding topic membership for all studies. In our case, topic membership $(1, \ldots, g)$ for a statistical methods section is inferred from the maximum coefficient value in the corresponding row of *H*, also known as the topic coherence score. For our two datasets, we applied NMF with *g* = 10 topics.

**Content analysis.**   Results were summarized by word clouds and n-gram analysis to identify frequently occurring terms within topics. Evidence of boilerplate text was assessed at the section and sentences levels using a modified version of the Jaccard similarity index. We chose the Jaccard index as an easy to interpret measure; for two pieces of tokenized text *A* and *B*, we defined the similarity score as $J(A, B) = |A \cap B|/|B|$. Calculating similarities relative to a target piece of text (*B*) allowed us to identify instances of similar text either as a complete sentence, or embedded within larger sentences. Analyses considered text tokenised at the word level, with locality-sensitive hashing applied to reduce the number of pairwise comparisons [37]. Instances of boilerplate text were defined by a Jaccard index of 0.9 or higher.

## Results

### Public Library of Science (PLOS ONE)

Targeted keyword searches using the *PLOS ONE* application programming interface (API) returned 131,847 papers, of which 111,731 (85%) included a statistical methods section (S1 Fig). In the final sample, 94,608 (85%) papers returned an exact match against one or more common section headings: 63,982 for 'statistical analysis', 13,343 for 'statistical analyses' and 13,510 for 'data analysis'. All papers included "Biology and life sciences" (n = 107,584), "Earth sciences" (n = 7,605) and/or "Computer and information sciences" (n = 5,190) in their top 3 subject classifications.

Statistical methods sections had a median length of 129 words and inter-quartile range of 63 to 258 words. 7,701 articles (7%) had a statistical methods section of 500 words or more.

19,077 articles (17%) had statistical methods sections with 50 words or less, equal to the length of this paragraph.

Topics reflected the use of statistical software (topics 3 and 5), descriptive statistics (topic 6), group based hypothesis testing (topics 1 and 4) and statistical significance (topics 1 and 9) (Fig 1). Also identified were topics related to regression (topic 2), meta-analysis (topic 7) and experimental designs (topic 10). At the section level, 528 studies (0.47%) were a direct cut-and-paste from another paper; 37,333 studies (33%) included at least one exact match at the sentence level.

Definitions of statistical significance at $\alpha = 0.05$ were the most common form of boilerplate text, found in approximately 1 in 10 of all included studies (Table 1). Topic 1 (n = 3,775) combined statistical significance with Student's t-test. Topic 9 (n = 6,104) focused on multiple thresholds for declaring statistical significance such as "$^*p < 0.05$, $^{**}p < 0.01$ and $^{***}p < 0.001$", a practice that has been criticized [38]. Minor variations of this phrase were identified in 40% of all studies assigned to this topic.

Statistical software topics differentiated between GraphPad Prism (topic 3: n = 9,879) and SPSS (topic 5: n = 9,574). Targeted searches for the n-gram "GraphPad Prism" returned 6,844 potential matches, including 263 studies that used the boilerplate text "statistical analysis was performed using GraphPad Prism" (Table 1). Common variants included software version (e.g. "version 5.0 for windows") and location information (e.g."La Jollie/San Diego CA USA"). Similar instances were identified for "SPSS" in topic 5, with 539 out of 9,005 studies (6%) identified as boilerplate text. Software details in both topics were frequently paired with hypothesis testing methods and definitions of statistical significance (S2 Fig).

Boilerplate text for descriptive statistics reflected the presentation of data as means plus or minus standard errors or standard deviations (Topic 6: 321/4,746 studies; 6.7%). In topic 2, an example of recycled text was "Continuous variables were expressed as mean ± standard deviation" (494 studies; 2.5%). Similar to other topics, descriptions were often paired with univariate hypothesis tests followed by more complex analyses, software and statements of statistical significance (S3 Fig).

## Australia and New Zealand Clinical Trials Registry (ANZCTR)

We downloaded 28,008 studies and found that 9,523 (34%) had a completed statistical methods section (S1 Fig). The median length of sections was 136 words with an inter-quartile range of 74 to 230 words. Eight studies were only one word, including "ANOVA", "SPSS" and even "SSPS".

Observational studies were less likely to have a missing statistical methods section compared with interventional studies (Table 2). Missing sections became less likely over time. Studies with more funders and a larger target sample size were less likely to have a missing statistical methods section.

Since studies registered with ANZCTR described planned analyses, we hypothesized that some studies did not specify statistical methods because they had yet to consult with a statistician. Targeted searches for "statistician" across all topics returned 381 studies, with examples including "Statistical analysis will be done in collaboration with a statistician" and "Pilot study at this point will use a statistician professionally to determine sample size calculations as required".

Topic modelling results reflected sample size calculations (topics 2 and 5), study designs (topics 4, 5, 6 and 8), quantitative methods (topics 3, 7, 9 and 10) and qualitative methods (topic 1) (Fig 2).

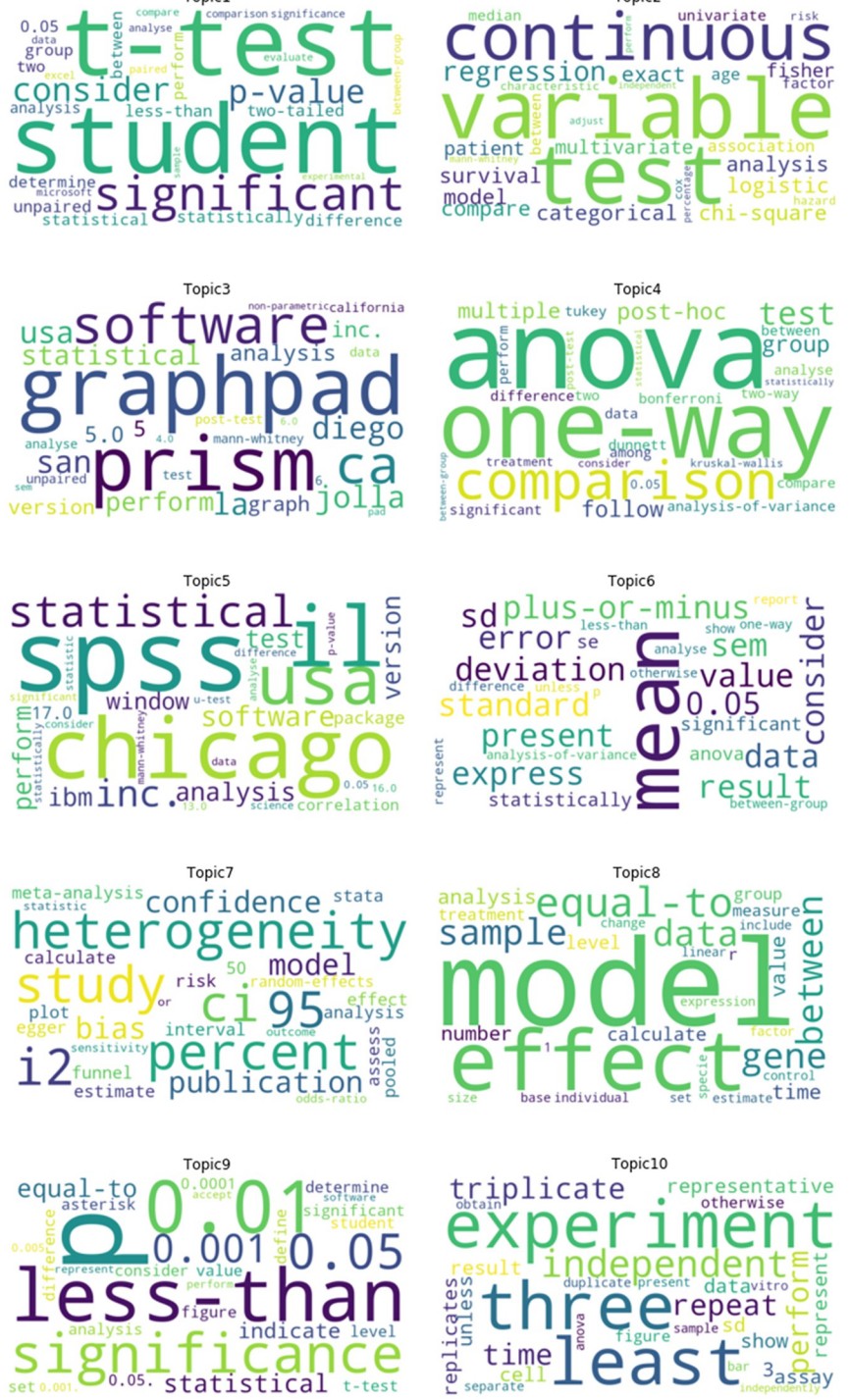

**Fig 1. Word clouds for ten topics for statistical methods sections published in PLOS ONE.**

Evaluation of boilerplate text revealed sections from 484 studies (5.1%) were close matches and 251 (2.6%) were an exact cut-and-paste from another study (Table 3). At the sentence level, the proportion of studies with shared text varied by topic, from 12% in topic 5 (pilot studies) to 38% in topic 3 (student's t-test) (S4 Fig).

**Table 1. Examples of boilerplate text from PLOS ONE papers based on targeted n-gram searches (sentence level).**

| Topic | Statistical methods text | Potential matches | Jaccard score | |
|---|---|---|---|---|
| | | | Median (IQR) | Boilerplate |
| 1 | Statistical analysis was performed using **student t-test** | 3,015 | 0.5 (0.5 to 0.75) | 189 |
| 2 | **Continuous variables** were expressed as mean **plus-or-minus** standard deviation | 1,228 | 0.82 (0.73 to 0.91) | 494 |
| | Categorical variables were expressed as **frequencies and percentages** | 643 | 0.75 (0.63 to 0.88) | 38 |
| 3 | All statistical analysis was performed using **Graphpad Prism** software | 6,844 | 0.56 (0.44 to 0.78) | 263 |
| 4 | **One-way** analysis of variance (**ANOVA**) was used for multiple comparisons and a Tukey post-hoc test was applied where appropriate | 6,660 | 0.43 (0.33 to 0.52) | 6 |
| 5 | Statistical analysis was performed using **SPSS** version 17.0 SPSS Inc Chicago IL USA | 9,005 | 0.58 (0.42 to 0.75) | 539 |
| 6 | Data are expressed as mean **plus-or-minus** SEM | 4,455 | 0.78 (0.67 to 0.89) | 321 |
| 7 | Summary estimates including **95 percent** confidence intervals (CIs) were calculated | 4,057 | 0.4 (0.3 to 0.5) | 6 |
| 8 | The significance level was set at p **equal-to 0.05** | 3,397 | 0.5 (0.5 to 0.7) | 262 |
| 9 | *p **less-than** 0.05 **p **less-than** 0.01 ***p **less-than** 0.001 | 5,559 | 0.83 (0.83 to 0.92) | 2,510 |
| 10 | All data are representative of at least three **independent experiments** | 1,722 | 0.6 (0.47 to 0.7) | 83 |
| All topics | A p-value **less-than** 0.05 was considered statistically significant | 64,639 | 0.6 (0.5 to 0.8) | 12,675 |
| | Data are presented as mean **plus-or-minus** SEM | 33,471 | 0.67 (0.67 to 0.78) | 1,648 |
| | Statistical analysis was performed using Student's **t-test** | 44,699 | 0.5 (0.38 to 0.63) | 1,043 |

N-grams are marked in bold. Potential matches refers to the number of studies that contained the target n-gram at least once. Boilerplate text was defined by a Jaccard score of 0.9 or higher. IQR: Inter-quartile range.

Thematic analysis of n-grams differentiated between study designs and statistical methods topics (S5 Fig). At the n-gram level, we noted the use of similar methods across multiple topics. For example, while topic 3 (student's t-test) was dominated by mentions of group-based hypothesis tests as expected, the same topic also referenced linear modelling/regression methods and descriptive statistics. Similarly, the use of linear modelling/regression methods was referenced across multiple topics covering quantitative and qualitative methods. Among study design topics, matching sentences highlighted the planned use of intention-to-treat analysis and descriptive statistics. For topic 6 (safety and tolerability studies), approximately 1 in 3 studies had evidence of boilerplate text at the sentence level, which included different combinations of summary statistics for presenting study variables. In contrast, topic 4 (efficacy and safety studies) returned 211 matches against the n-gram "95 percent"; subsequent review

**Table 2. Logistic regression results for study characteristics associated with missing statistical methods sections in ANZCTR.**

| Variable | Odds ratio | 95% CI |
|---|---|---|
| Study type = Observational | 0.78 | (0.69, 0.89) |
| Date (per year) | 0.90 | (0.88, 0.91) |
| Number of funders | 0.80 | (0.74, 0.86) |
| Target sample size (per doubling) | 0.90 | (0.88, 0.92) |

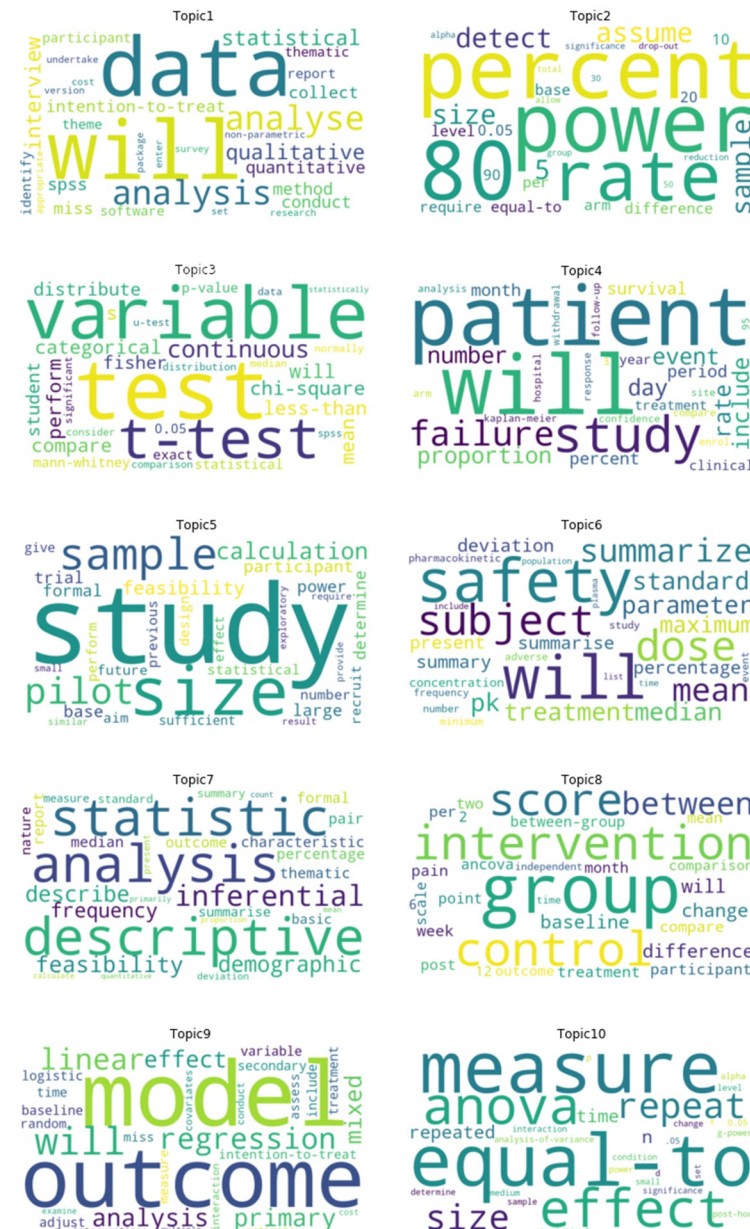

**Fig 2. Word clouds for ten topics for statistical methods sections published in ANZCTR.**

identified 28 studies that were close matches to be phrase "at a confidence level of 95% and a precision around the estimate of 5%, a minimum of 73 patients will be included". Among methods topics, definitions of statistical significance was a recurring theme. Some topics simply stated the main analysis method, for example, "descriptive statistics" (topic 7; 16 exact matches). Examples of close matching sentences and Jaccard similarity scores are given in Table 4. Finally, we noted the use of the same methods among subgroups of topics. For example,

**Table 3. Results of boilerplate analysis applied to the ANZCTR dataset.**

| Topic (Number of studies) | Word count Median (IQR) | Sentences Median (IQR) | Matching studies | |
|---|---|---|---|---|
| | | | Section | 1+ sentences |
| 1: Qualitative methods (842) | 116 (58 to 207) | 6 (3 to 10) | 46 (23) | 196 (171) |
| 2: Sample size calculations (1,753) | 147 (92 to 231) | 6 (3 to 9) | 40 (22) | 311 (259) |
| 3: Student's t-test (923) | 119 (75 to 178) | 6 (4 to 8) | 62 (32) | 354 (292) |
| 4: Efficacy and safety studies (871) | 174 (97 to 268) | 7 (4 to 12) | 56 (7) | 190 (162) |
| 5: Pilot studies (737) | 78 (40 to 129) | 4 (2 to 6) | 39 (24) | 88 (78) |
| 6: Safety and tolerability studies (507) | 127 (73 to 220) | 6 (4 to 10) | 40 (23) | 182 (159) |
| 7: Descriptive analysis (328) | 39 (20 to 65) | 2 (1 to 4) | 43 (41) | 59 (57) |
| 8: Intervention studies (826) | 174 (98 to 275) | 7 (4 to 11) | 14 (6) | 129 (106) |
| 9: Linear models (1,728) | 172 (95 to 298) | 7 (4 to 12) | 85 (44) | 554 (486) |
| 10: Analysis of variance (1,008) | 131 (76 to 214) | 5 (3 to 9) | 59 (29) | 236 (209) |

The number of studies with Jaccard similarity scores greater than or equal to 0.9 from pairwise comparisons are presented; the number of studies with cut-and-pasted text is given in brackets.

## Discussion

The aim of our analysis was to identify common themes in statistical methods sections both in terms of chosen methods and how these methods are being communicated. Our findings provide evidence of boilerplate statistical methods sections, resulting from likely cut-and-pasting and slight modifications to existing text descriptions. Results from topic modeling further identified distinct themes across statistical methods sections that emphasised details about study design, chosen methods, p-values and software. This is a strong sign of the ritualistic practice of statistics where researchers go through the motions rather than using conscientious practice [2].

Despite the extensive array of statistical tests available, our results show that authors are often reporting the same few methods. In related work, a content-based analysis of ecology and conservation journals summarised trends in linear modelling using n-grams including "t-test", "ANOVA" and "regression"; results provided evidence of a movement towards model-based inference [39]. We found that Student's t-test and ANOVA were commonly cited methods for comparing groups in both *PLOS ONE* and ANZCTR datasets. For statistical methods sections in *PLOS ONE*, we also found that many studies followed a generic template, combining chosen statistical methods with descriptive statistics for summarizing data, statements of statistical significance and/or choice of software. When investigating cases of boilerplate text, results based on n-grams versus close matches at the sentence level varied considerably by topic. These findings suggest that there is a tendency for researchers to default to the same common statistical methods when completing analyses, in line with the view of statistical analysis as a mechanistic process. However, for studies that use the same statistical methods, text used to describe important details may vary.

Defining statistical significance at $p < 0.05$ was the most common example of boilerplate text in both datasets. The widespread use of statistical significance is troubling given the bright-line thinking it engenders [40] and the common misinterpretations of p-values [41, 42]. Nonetheless, conflicting views about the use of statistical significance remain. In a follow-up survey of signatories to an article calling for the end of statistical significance [43], 22% of respondents said they were likely to continue using the concept in future publications [44]. Reasons cited included the mindful use of statistical significance in combination with other evidence and concerns about the feasibility of abandoning statistical significance given its

**Table 4. Example boilerplate text from ANZCTR studies with the highest number of matches per topic (sentence level).**

| Topic | Statistical methods text | Potential matches | Jaccard score | |
|---|---|---|---|---|
| | | | Median (IQR) | Boilerplate |
| 1 | All analyses will be conducted on an **intention-to-treat** basis | 153 | 0.55 (0.52 to 0.73) | 11 |
| 2 | The **sample size** is adjusted for a 10% drop-out rate | 1,224 | 0.42 (0.33 to 0.5) | 9 |
| 3 | Continuous normally distributed variables will be compared using **student t-test** and reported as means standard deviation while non-normally distributed data will be compared using wilcoxon rank-sum tests and reported as medians inter-quartile range | 134 | 0.32 (0.2 to 0.4) | 8 |
| 4 | At a confidence level of **95 percent** and a precision around the estimate of 5% a minimum of 73 patients will be included | 211 | 0.46 (0.33 to 0.58) | 28 |
| 5 | No formal **sample size calculation** was performed | 163 | 0.43 (0.43 to 0.57) | 4 |
| 6 | Continuous variables will be summarized by mean standard deviation **median minimum and maximum** | 65 | 0.77 (0.69 to 0.85) | 15 |
| 7 | **Descriptive statistics** will be used | 246 | 0.8 (0.55 to 0.8) | 69 |
| 8 | Analyses will be conducted on an **intention-to-treat** basis | 149 | 0.55 (0.46 to 0.73) | 16 |
| 9 | **Linear mixed models** will be used to analyze the data | 238 | 0.6 (0.6 to 0.8) | 20 |
| 10 | Data will be analyzed using standardised non-parametric or parametric statistical methods where appropriate (using) **repeated measures ANOVA** | 206 | 0.29 (0.24 to 0.35) | 5 |
| All topics | A p-value **less-than** 0.05 will be considered statistically significant | 1,967 | 0.55 (0.36 to 0.73) | 267 |
| | Analyses will be conducted on an **intention-to-treat** basis | 1,630 | 0.6 (0.5 to 0.7) | 191 |
| | Baseline characteristics will be summarised using **descriptive statistics** | 1,375 | 0.5 (0.5 to 0.63) | 23 |

The number of matching to each sentence was based on a Jaccard score of 0.9 or higher. Potential matches refers to the number of studies that contained the target n-gram at least once.

engrained usage in published literature. At the same time, null hypothesis significance testing has been cited as a root cause fueling the reproducibility crisis, and a problem that has been difficult to shift [45].

Two topics identified in the *PLOS ONE* dataset highlighted statistical software. Similarly, some sections extracted from ANZCTR only stated the software, implying that this was the primary criterion for statistical analysis. As Doug Altman said, "Many people think that all you need to do statistics is a computer and appropriate software" [6]. This is far from the truth, and whilst it is important for researchers to mention the software and version used for reproducibility purposes, it is a minor detail compared with explaining which methods were used and why.

One reason inadequate methods sections get published is because many journals do not use statistical reviewers, despite empirical evidence showing they improve manuscript quality [12]. It is possible that the exact details of statistical methods are viewed as relatively unimportant by authors and reviewers, and something that can be read last or even skipped [46]. Some journals foster this lack of importance by putting the methods section last. Statistical methods sections may be getting less scrutiny than other sections both because of their position in the paper, relatively low word counts, and because they so often contain boilerplate text. Another potential reason is that authors resort to boilerplate text is because of the overly-critical approach to statistics by some reviewers who pounce on anything outside the accepted dogma [47].

Whilst checklists are a useful tool to improve statistical reporting, peer review by nonstatistical reviewers and editors cannot replace expert appraisal on the appropriateness of statistical methods used [48]. Mechanisms to encourage authors to share their analysis code would provide an alternative route for checking what statistical methods were used. This is not a perfect solution, as we still want authors to accurately report their methods in their paper, but it does increase transparency. A recent paper found that code sharing was very low in biomedical papers, with just 2% of a sample of over 6,000 papers sharing code [49]. The introduction of incentives for code sharing such as article badges has to date shown limited efficacy [50], however further research in this area may offer potential solutions for promoting reproducibility.

Our approach for identifying boilerplate text was not intended as a form of plagarism detection, but rather as evidence of standardised descriptions being used. For simple study designs, a boilerplate description might be adequate to promote consistency in reporting and meet reporting requirements. For example, ANZCTR sections commonly reported sample size justifications and planned analyses using intention-to-treat principles. Beyond statistical methods sections, initiatives such as 2WeekSR have been developed to streamline the completion of systematic reviews, including the use of automation to generate consistent descriptions of results suitable for using in papers [51]. However, if boilerplate descriptions are to be used, they must provide readers with sufficient details to confirm that appropriate methods were used and enable independent verification of results. Unfortunately, this is not always the case. For example, a study of papers that used ANOVA found 95% did not contain the information needed to determine what type of ANOVA was performed. This lack of information could well be because the authors used a boilerplate statistical methods section that was missing key details.

Our analysis focused on studies with a clearly marked statistical methods section, based on predefined section headings. It is therefore possible that some of the papers excluded from our analysis conducted statistical analyses but placed descriptions elsewhere. For *PLOS ONE*, excluded papers may have described statistical methods as part of the supplementary material, which tend to be less structured than the main text. Similarly, since submissions to both PLOS ONE and ANZCTR do not undergo compulsory statistical review, our results may not be generalizable to all journals and registries, especially those that consistently use a statistical reviewer. Given the large sample sizes for both datasets, it was not feasible to check whether papers used the correct methods.

## Supporting information

**S1 Fig. Inclusion flowchart for studies downloaded for each case study.** A: *PLOS ONE*; B: ANZCTR. For (A), "Non-specific analysis" refers to studies where the use of statistical methods could not be determined by on section headings; e.g., "Microarray analysis".
(TIF)

**S2 Fig. Common combinations of statistical methods in topic related to the use of GraphPad Prism (topic 3) and SPSS (topic 5).** General themes for statistical methods were based on targeted word searches and categorized into statistical significance, descriptive statistics, parametric hypothesis tests, nonparametric hypothesis tests, linear modelling/regression and software. The most frequent combinations of themes are given on the x-axis, with the corresponding number of studies on the y-axis.
(TIF)

**S3 Fig. Common combinations of statistical methods in topic related to descriptive statistics (topics 2 and 6).** General themes for statistical methods were based on targeted word searches and categorized into statistical significance, descriptive statistics, parametric

hypothesis tests, nonparametric hypothesis tests, linear modelling/regression and software. The most frequent combinations of themes are given on the x-axis, with the corresponding number of studies on the y-axis.
(TIF)

**S4 Fig. Total matching sentences by topic for the ANZCTR dataset.** A match was defined any pair of sentences between ANZCTR studies with a Jaccard score equal to 0.9 or higher.
(TIF)

**S5 Fig. Summary of close matches at the sentence level (x-axis) by ANZCTR themes inferred from common n-grams (y-axis), organized by study design (A) and methods-based (B) topics.** A close match was defined any pair of sentences between ANZCTR studies with a Jaccard score equal to 0.9 or higher.
(TIF)

## Acknowledgments

The authors gratefully acknowledge computational resources and services used in this work provided by the eResearch Office, Queensland University of Technology, Brisbane, Australia. We thank Lee Jones for constructive feedback on the draft manuscript.

## Author Contributions

**Conceptualization:** Nicole M. White, Thirunavukarasu Balasubramaniam, Richi Nayak, Adrian G. Barnett.

**Data curation:** Nicole M. White, Adrian G. Barnett.

**Formal analysis:** Nicole M. White, Thirunavukarasu Balasubramaniam, Richi Nayak, Adrian G. Barnett.

**Writing – original draft:** Nicole M. White, Thirunavukarasu Balasubramaniam, Richi Nayak, Adrian G. Barnett.

**Writing – review & editing:** Nicole M. White, Thirunavukarasu Balasubramaniam, Richi Nayak, Adrian G. Barnett.

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
