## [Decision Letter · Decision Letter 0]

20 Dec 2021

PONE-D-21-33560An observational analysis of the trope "A p-value of < 0.05 was considered statistically significant" and other cut-and-paste statistical methodsPLOS ONE

Dear Dr. White,

Thank you for submitting your manuscript to PLOS ONE. After careful consideration, we feel that it has merit but does not fully meet PLOS ONE’s publication criteria as it currently stands. Therefore, we invite you to submit a revised version of the manuscript that addresses the points raised during the review process.

Kindly address all comments from reviewers 1 and 2. Reviewer 1 has suggested some additional literature to include in your discussion, I would be very grateful if you could consider these suggestions. 

We look forward to receiving your revised manuscript.

Kind regards,

Benedikt Ley, PhD

Academic Editor

PLOS ONE

Journal Requirements:

2. We note that Figure 1 and 2 in your submission contain copyrighted images. All PLOS content is published under the Creative Commons Attribution License (CC BY 4.0), which means that the manuscript, images, and Supporting Information files will be freely available online, and any third party is permitted to access, download, copy, distribute, and use these materials in any way, even commercially, with proper attribution. For more information, see our copyright guidelines: http://journals.plos.org/plosone/s/licenses-and-copyright.

 a. You may seek permission from the original copyright holder of Figure 1 and 2 to publish the content specifically under the CC BY 4.0 license.

Reviewers' comments:

Reviewer's Responses to Questions

**Comments to the Author**

1. Is the manuscript technically sound, and do the data support the conclusions?

Reviewer #1: Yes

Reviewer #2: Yes

2. Has the statistical analysis been performed appropriately and rigorously? 

Reviewer #1: Yes

Reviewer #2: Yes

3. Have the authors made all data underlying the findings in their manuscript fully available?

Reviewer #1: Yes

Reviewer #2: Yes

4. Is the manuscript presented in an intelligible fashion and written in standard English?

Reviewer #1: Yes

Reviewer #2: Yes

5. Review Comments to the Author

Reviewer #1: I think the authors of this paper have made a commendable effort to screen two major public scientific databases. I have two major comments, one of which is somewhat personal.

1) The paper is at least partly framed as if the main problem is that sentences are copied from other papers; at least this was my first impression from the abstract. However, I think one could well argue that for the sake of clarity and comprehensibility, the same words *should* be used by different authors for the same methods. Thus, if a study used the default alpha of 0.05, a sentence like “a p-value of < 0.05 was considered statistically significant” *should* be written in the paper. Given that probably most screened studies did use the default 0.05, I think it is actually a bad sign that only 1 in 10 studies explicitly wrote it.

If we want that people become aware that they are using the same statistical default options since almost a century, it would be helpful if they wrote in their papers, e.g., “we used the default p = 0.05 significance level and the default null hypothesis of zero effect”.

I thus suggest you should clarify and discuss that your aim was not to study plagiarism but the extent to which people use the same old default methods – or at least the extent to which they explicitly write that they used those methods, which is probably not the same thing, which you should discuss.

I suggest that your results on the use of boilerplate text strongly underestimate the extent of a ritualistic practice of statistics, and that it would be desirable if more people would clearly say that they are committed to a ritualistic practice. In other words, I suggest that more use of boilerplate text might be desirable if people use boilerplate methods.

2) You did not extensively repeat the discussion about statistical significance, which is fine. However, I happen to be author of the article calling for the end of statistical significance that you discussed but did not cite (here’s the article: https://doi.org/10.1038/d41586-019-00857-9). I thus have three minor comments on your discussion starting from line 269:

a) You cited the paper by Diaz-Quijano et al. (2020) who performed a survey on our signatories and found that around 22% of the 151 respondents said that they would likely use the concept of “statistical significance” again in future publications. I think you should write “22% of respondents *said they* were likely to continue using the concept in future publications”.

b) You then wrote that “Reasons cited included the mindful use of p-values in combination with other evidence and concerns about the feasibility of abandoning p-values given their engrained usage in published literature.”

You should exchange both mentions of “p-values” with “statistical significance”, as is written in Diaz-Quijano et al. (2020) and in our original article “Retire statistical significance”. Of course, statistical significance and P-values are not the same. I and my co-authors have nothing against mindful use of P-values, which is what we try to advocate. On a side note, I would probably be among the people saying they would likely use the concept of statistical significance in future publications; for example, I’m using it whenever I write about the problems with this concept.

c) Your reference for misinterpretations of P-values is Goodman’s “A Dirty Dozen”, which is a good paper; an updated version that you could cite as well and on which Goodman is co-author is https://doi.org/10.1007/s10654-016-0149-3.

Signed review: Valentin Amrhein

Reviewer #2: This is an interesting paper which aims to quantify the extent of ‘cutting and pasting’ or ‘recycling’ of statistical methods sections across publications and trial registries. This is an attempt to quantify and better understand poor reporting of statistical methods and statistical methods used in published papers and registered trials. The authors included statistical methods sections from articles published in PLOS ONE and study protocols registered in the Australian and New Zealand Clinical Trials Registry. This was an interesting paper with unsurprising but novel findings. Poor statistical methods is an ongoing problem and this is highlighted by the authors’ finding that only 17% of articles had a statistical methods section with 50 words or less! I have only minor comments/suggestions to the manuscript, which are provided below. The authors should also be commended for making all of their code freely available.

Minor comments:

Methods:

1. I note that the authors didn’t include a heading of ‘statistical methods’ in their manuscript, which of course means that their own study wouldn’t be picked up in future update of this study.

2. In Steps 1 and 2 for searching PLOS ONE, it would be helpful to provide the link to the code in Github.

3. The last paragraph of the Data sources/ANZCTR section describes a statistical analysis to investigate if particular studies were more likely to be missing a statistical methods section. This section doesn’t seem to fit with the rest of the section as it describes a statistical method (specific to the ANZCTR).

Results:

4. In Figure S1, it’s not clear what ‘non-specific analysis’ means.

5. In lines 193-195, the reported figures (95518, 64144, 13380 and 13627) don’t match the figures in S1. It would be good if there was a more obvious link between the text and the figure.

6. Similar to the above comment, in lines 230, the figure ‘9623 had a completed statistical methods section’ doesn’t match the `analysed’ figure in Fig S1. It would be good if there was an obvious link between the text and the figure.

6. PLOS authors have the option to publish the peer review history of their article (what does this mean?). If published, this will include your full peer review and any attached files.

Reviewer #1: **Yes: **Valentin Amrhein

Reviewer #2: No

---

## [Author Response · Author response to Decision Letter 0]

6 Feb 2022

The editor asked us to consider some additional literature to include in the discussion, suggested by Reviewer 1. All of these suggestions have been incorporated in the revised manuscript.

Reviewer #1: I think the authors of this paper have made a commendable effort to screen two major public scientific databases. I have two major comments, one of which is somewhat personal.

1) The paper is at least partly framed as if the main problem is that sentences are copied from other papers; at least this was my first impression from the abstract. However, I think one could well argue that for the sake of clarity and comprehensibility, the same words *should* be used by different authors for the same methods. Thus, if a study used the default alpha of 0.05, a sentence like “a p-value of < 0.05 was considered statistically significant” *should* be written in the paper. Given that probably most screened studies did use the default 0.05, I think it is actually a bad sign that only 1 in 10 studies explicitly wrote it.

If we want that people become aware that they are using the same statistical default options since almost a century, it would be helpful if they wrote in their papers, e.g., “we used the default p = 0.05 significance level and the default null hypothesis of zero effect”.

I thus suggest you should clarify and discuss that your aim was not to study plagiarism but the extent to which people use the same old default methods – or at least the extent to which they explicitly write that they used those methods, which is probably not the same thing, which you should discuss.

I suggest that your results on the use of boilerplate text strongly underestimate the extent of a ritualistic practice of statistics, and that it would be desirable if more people would clearly say that they are committed to a ritualistic practice. In other words, I suggest that more use of boilerplate text might be desirable if people use boilerplate methods.

RESPONSE: Thank you for offering this interesting perspective. We agree that boilerplate text may be beneficial to provide clarity about the same method and/or key assumptions made as part of analysis. There may also be cases where a boilerplate section may be adequate to meet reporting requirements, provided that sufficient details are given to allow readers to assess study quality. We alluded to this point in the original manuscript (Discussion, p9 lines 296 – 299), but agree that this discussion could be expanded. We hope that our paper will reinforce the statistical community’s concern around the ritualistic practice of statistics and poor reporting. 

To address this feedback, we have added new text throughout the manuscript to clarify the purpose of our analysis as follows:

Abstract: the opening sentence of the second paragraph has been revised to “Instances of boilerplate text suggest a mechanistic approach to statistical analysis, where the same default methods are being used and described using standardized text. To investigate the extent of this practice, we analyzed text extracted from published statistical methods sections from PLOS ONE and the Australian and New Zealand Clinical Trials Registry (ANZCTR).”

Introduction, page 2 

-Line 33: The opening sentence has been modified to “A potential contributor to poor reporting is the temptation for researchers to re-use descriptions of the same default statistical methods, to make their papers resemble those of their peers and increase perceived chances of publication [17]. As these default choices become more common, valid criticism by reviewers and journal editors becomes increasingly difficult, as past use may be argued by researchers as offering precedent for the conduct of analysis within their discipline [18]”

-Line 46: the final sentence of the Introduction has been revised to “The use of boilerplate text indicates that researchers are emphasizing the same details about chosen statistical analyses, and potentially giving little thought into the conduct and transparent reporting of statistical methods used.”

Results: ANZCTR, page 9

-lines 258 – 263: Additional text has been added to highlight the use of the same statistical methods across topics, as reflected in Figure S5 “At the n-gram level, we noted the use of similar methods across multiple topics. For example, while topic 3 (student's t-test) was dominated by mentions of group-based hypothesis tests as expected, the same topic also referenced linear modelling/regression methods and descriptive statistics. Similarly, the use of linear modelling/regression methods was referenced across multiple topics covering quantitative and qualitative methods.”. This finding is somewhat expected as statistical methods sections can either state a single method for analysis, or multiple related methods as part of the same analysis.

Discussion, pages 8 – 11

-line 277: Added the opening sentence “The aim of our analysis was to identify common themes in statistical methods sections both in terms of chosen methods and how these methods are being communicated”.

-line 280: Added the sentence “Results from topic modeling further identified distinct themes across statistical methods sections that emphasized details about study design, chosen methods, p-values and software”

-page 9, lines 285 – 299: This paragraph has been completely revised to emphasize the finding that researchers tend to use the same statistical methods repeatedly, although the text used to describe them may vary. In the original manuscript, this paragraph was the fourth paragraph of the Discussion. In the revised manuscript, this paragraph has been moved up to the second paragraph of the Discussion given its focus on the use of the same methods across studies. 

-page 10, lines 339 – 353: This is a new paragraph incorporating text from the original Discussion section. The paragraph expands on the discussion of cases when boilerplate text might be sufficient for consistency/to meet reporting requirements. In this paragraph, we suggest cases where boilerplate text may be helpful but emphasise that such text must contain sufficient details to enable independent replication of results. As part of this discussion, we provide an example of an automation initiative (2WeekSR) for assisting with completion of systematic reviews. 

2) You did not extensively repeat the discussion about statistical significance, which is fine. However, I happen to be author of the article calling for the end of statistical significance that you discussed but did not cite (here’s the article: https://doi.org/10.1038/d41586-019-00857-9). I thus have three minor comments on your discussion starting from line 269:

a) You cited the paper by Diaz-Quijano et al. (2020) who performed a survey on our signatories and found that around 22% of the 151 respondents said that they would likely use the concept of “statistical significance” again in future publications. I think you should write “22% of respondents *said they* were likely to continue using the concept in future publications”.

b) You then wrote that “Reasons cited included the mindful use of p-values in combination with other evidence and concerns about the feasibility of abandoning p-values given their engrained usage in published literature.”

You should exchange both mentions of “p-values” with “statistical significance”, as is written in Diaz-Quijano et al. (2020) and in our original article “Retire statistical significance”. Of course, statistical significance and P-values are not the same. I and my co-authors have nothing against mindful use of P-values, which is what we try to advocate. On a side note, I would probably be among the people saying they would likely use the concept of statistical significance in future publications; for example, I’m using it whenever I write about the problems with this concept.

c) Your reference for misinterpretations of P-values is Goodman’s “A Dirty Dozen”, which is a good paper; an updated version that you could cite as well and on which Goodman is co-author is https://doi.org/10.1007/s10654-016-0149-3.

RESPONSE: Thank you for bringing this missing reference to our attention. We have now cited this article in the revised Discussion (page 10, reference [43]). Further revisions have been completed as follows:

For (a) and (b), the relevant text has been updated as suggested.

For (c) The suggested reference has been added to the Discussion (page 10, reference [42]).

Reviewer #2: This is an interesting paper which aims to quantify the extent of ‘cutting and pasting’ or ‘recycling’ of statistical methods sections across publications and trial registries. This is an attempt to quantify and better understand poor reporting of statistical methods and statistical methods used in published papers and registered trials. The authors included statistical methods sections from articles published in PLOS ONE and study protocols registered in the Australian and New Zealand Clinical Trials Registry. This was an interesting paper with unsurprising but novel findings. Poor statistical methods is an ongoing problem and this is highlighted by the authors’ finding that only 17% of articles had a statistical methods section with 50 words or less! I have only minor comments/suggestions to the manuscript, which are provided below. The authors should also be commended for making all of their code freely available.

Minor comments:

Methods:

1. I note that the authors didn’t include a heading of ‘statistical methods’ in their manuscript, which of course means that their own study wouldn’t be picked up in future update of this study.

RESPONSE: Thank you for identifying this oversight! We have added the heading ‘Statistical methods’ to the Materials and Methods section of the revised manuscript; see page 4, line 120. We have also updated Figure S1 to include the number of studies that included either “Statistical methods” or “Statistical methodology” as a section heading. Both section headings were included in our initial analysis as partial matches to “Statistical method”, but we have included for clarity.

2. In Steps 1 and 2 for searching PLOS ONE, it would be helpful to provide the link to the code in Github.

RESPONSE: We have added the following sentence after the explanation of Steps 1 and 2 for searching PLOS ONE (page 3, line 94)

“Code to complete steps 1 and 2 is available at https://git/hub.com/agbarnett/stats_section/code/plosone.”

To improve the organisation of the GitHub repository, we have further updated the README file and folder structure for the cited GitHub repository has been updated for clarity. We hope that these changes make our code more accessible to readers.

3. The last paragraph of the Data sources/ANZCTR section describes a statistical analysis to investigate if particular studies were more likely to be missing a statistical methods section. This section doesn’t seem to fit with the rest of the section as it describes a statistical method (specific to the ANZCTR).

RESPONSE: We agree that this paragraph would be better placed in the Statistical Methods section. In the revised manuscript, we have moved this paragraph to the Statistical Methods section of the revised manuscript and added the subsection heading ‘Analysis of missing statistical methods sections’ (see page 4, line 141 – 150) The revised text is:

“Statistical methods sections were missing for some studies downloaded from ANZCTR, including sections labelled as ``Not applicable'', ``Nil'' or ``None''. Since these studies would be excluded from topic modeling, we examined if there were particular studies where the statistical methods section was more likely to be missing.”

Results:

4. In Figure S1, it’s not clear what ‘non-specific analysis’ means.

RESPONSE: We used the term ‘non-specific analysis’ to describe studies that including the word ‘analysis’ as part of its section heading, but could not be matched against our set of predefined section headings; e.g., ‘Microarray analysis’. We agree that this could be communicated better and have amended the caption for Figure S1 to include a definition for non-specific analysis (page 11, line 359)

5. In lines 193-195, the reported figures (95518, 64144, 13380 and 13627) don’t match the figures in S1. It would be good if there was a more obvious link between the text and the figure.

RESPONSE: Thank you for pointing out this discrepancy. Figure S1 has been updated to include numbers reported in the main text. Please note that revised numbers in the manuscript have changed to resolve double counting of studies. For the PLOS ONE dataset, we have also included the exclusion reason “Other” as part of the flowchart, to flag remaining studies where initial search terms could not be attributed to standard section headings within XML files.

6. Similar to the above comment, in lines 230, the figure ‘9623 had a completed statistical methods section’ doesn’t match the `analysed’ figure in Fig S1. It would be good if there was an obvious link between the text and the figure.

RESPONSE: Figure S1 has been updated to match reported numbers in the main text. When revising, we noted a small error in the final sample size where 9,632 should be 9,523. This discrepancy has been resolved in the Abstract, the main text and in Tables 3 and 4.

A copy of reviewer responses has been uploaded as the file 'Response to Reviewers.pdf'

---

## [Editor Report · Decision Letter 1]

9 Feb 2022

An observational analysis of the trope "A p-value of < 0.05 was considered statistically significant" and other cut-and-paste statistical methods

PONE-D-21-33560R1

Dear Dr. White,

We’re pleased to inform you that your manuscript has been judged scientifically suitable for publication and will be formally accepted for publication once it meets all outstanding technical requirements.

Kind regards,

Benedikt Ley, PhD

Academic Editor

PLOS ONE

Additional Editor Comments (optional):

Many thanks for addressing all comments by the reviewers and commenting on the resolved coding issues. 
---

## [Editor Report · Acceptance letter]

15 Feb 2022

PONE-D-21-33560R1 

An observational analysis of the trope “A p-value of < 0.05 was considered statistically significant” and other cut-and-paste statistical methods 

Dear Dr. White:

I'm pleased to inform you that your manuscript has been deemed suitable for publication in PLOS ONE. Congratulations! Your manuscript is now with our production department. 

Kind regards, 

on behalf of

Dr Benedikt Ley 

Academic Editor

PLOS ONE